# GAL3ST1 Deficiency Reduces Epithelial–Mesenchymal Transition and Tumorigenic Capacity in a Cholangiocarcinoma Cell Line

**DOI:** 10.3390/ijms25137279

**Published:** 2024-07-02

**Authors:** Lin Chen, Montserrat Elizalde, Ludwig J. Dubois, Anjali A. Roeth, Ulf P. Neumann, Steven W. M. Olde Damink, Frank G. Schaap, Gloria Alvarez-Sola

**Affiliations:** 1Department of Surgery, School of Nutrition and Translational Research in Metabolism, Maastricht University, 6200 MD Maastricht, The Netherlands; linchen.med@gmail.com (L.C.); aroeth@ukaachen.de (A.A.R.); uneumann@ukaachen.de (U.P.N.); steven.oldedamink@maastrichtuniversity.nl (S.W.M.O.D.); frank.schaap@maastrichtuniversity.nl (F.G.S.); 2Division of Gastroenterology-Hepatology, Department of Internal Medicine, Maastricht University, 6200 MD Maastricht, The Netherlands; m.elizalde@maastrichtuniversity.nl; 3The M-Lab, Department of Precision Medicine, Maastricht University, 6200 MD Maastricht, The Netherlands; ludwig.dubois@maastrichtuniversity.nl; 4Department of General, Visceral and Transplant Surgery, University Hospital Aachen, 52074 Aachen, Germany

**Keywords:** cholangiocarcinoma, glycolysis, sulfatides, *GAL3ST1*, epithelial–mesenchymal transition, barrier function, cancer therapeutic target

## Abstract

Cholangiocarcinoma (CCA), or bile duct cancer, is the second most common liver malignancy, with an increasing incidence in Western countries. The lack of effective treatments associated with the absence of early symptoms highlights the need to search for new therapeutic targets for CCA. Sulfatides (STs), a type of sulfoglycosphingolipids, have been found in the biliary tract, with increased levels in CCA and other types of cancer. STs are involved in protein trafficking and cell adhesion as part of the lipid rafts of the plasma membrane. We aimed to study the role of STs in CCA by the genetic targeting of GAL3ST1, an enzyme involved in ST synthesis. We used the CRISPR-Cas9 system to generate *GAL3ST1*-deficient TFK1 cells. *GAL3ST1* KO cells showed lower proliferation and clonogenic activity and reduced glycolytic activity compared to TFK1 cells. Polarized TFK1 *GAL3ST1* KO cells displayed increased transepithelial resistance and reduced permeability compared to TFK1 *wt* cells. The loss of GAL3ST1 showed a negative effect on growth in 30 out of 34 biliary tract cancer cell lines from the DepMap database. GAL3ST1 deficiency partially restored epithelial identity and barrier function and reduced proliferative activity in CCA cells. Sulfatide synthesis may provide a novel therapeutic target for CCA.

## 1. Introduction

Cholangiocarcinoma (CCA), or bile duct cancer, is a group of malignancies originating in the epithelium of the biliary tree [1]. CCA is a heterogeneous group of tumors, classified by anatomical origin as intrahepatic, perihilar, and distal CCA [2,3]. CCA is the second most common primary liver neoplasm, with an increasing incidence in Western countries [4]. Although primary liver cancer mortality is currently more uniform in Europe, it shows an overall reduction associated with hepatocellular carcinoma but a marked increase related to CCA [5].

The global incidence and mortality of CCA show geographical variations, being overall higher in Eastern countries. These differences may reflect genetic and environmental risk factors involved in CCA formation [6]. In recent years, an increase in global incidence and mortality has been observed, with a different epidemiological trend between the different CCA subtypes [7]. This increasing incidence due to the late diagnosis and lack of effective therapies compromises the treatment of CCA and highlights the need to search for new therapeutic targets that reduce tumor growth and metastatic potential.

CCA cells are characterized by the loss of cell cycle control due to the activation of oncogenes and inactivation of tumor suppressor genes, resulting in uncontrolled growth and the ability to spread to other organs and tissues [7]. CCA is an aggressive type of cancer with great invasive capacity; therefore, local tumor cells must acquire migratory capacity that allows them to spread to and invade other tissues [8]. Epithelial–mesenchymal transition (EMT) is one of the most studied mechanisms regarding metastatic capacity in CCA. During this process, the cells of the biliary epithelium (i.e., cholangiocytes) acquire mesenchymal characteristics required for cancer cells to migrate to and invade surrounding tissues [9].

The biliary tree consists of a system of interconnected vessels with increasing diameter, are lined with cholangiocytes, and transport bile formed by hepatocytes to the duodenum. Bile is a digestive solution produced and secreted by the liver, which also allows the elimination of waste products [10]. The bile duct epithelium shows secretory functions but also forms a barrier that prevents the entry of toxic substances from the bile into the liver parenchyma [11]. Defective blood–bile barrier function induces parenchymal cell damage by activating an inflammatory response that can contribute to the development of hepatobiliary diseases like primary sclerosing cholangitis (PSC), primary biliary cholangitis (PBC), and CCA [9].

The main treatment for intrahepatic cholangiocarcinoma (iCCA) is surgical resection; however, most patients present with unresectable tumors [12]. Liver transplantation is considered in the very early stages of iCCA (single tumors ≤ 2 cm), but there is debate on its widespread use due to the shortage of donated organs and the need for precise patient selection [13]. Larger tumor size is associated with a high rate of recurrence after liver transplantation [14]. When surgical resection or liver transplantation is not feasible, cytotoxic drugs such as cisplatin and gemcitabine are used, and these increase the median overall survival to 8–12 months, depending on individual or combined use [15].

Sulfatides (STs) are a class of sulfoglycosphingolipids present in the liver but are restricted to the bile ducts [16,17]. We recently demonstrated the presence of STs in iCCA tissue, although their levels were not different on a per-cell basis compared to tumor-distal tissue (control). A high ratio of unsaturated to saturated STs in iCCA tumor tissue was associated with early tumor recurrence [17]. STs are adhesive molecules present in the plasma membrane and are involved in the trafficking of proteins in different epithelial cells [18]. The presence of STs in the biliary tract may be part of a protective mechanism related to the blood–bile barrier function of this tissue. This possible protective role added to its function as an adherent molecule and its presence in iCCA, warranting the study of the function of these lipids in bile duct cancer. It is important to find new strategies that inhibit tumor growth and spread. For this purpose, we set out to evaluate the role of STs in CCA.

Our goal in this work was to study the role of endogenous sulfatides in a CCA cell line by targeting GAL3ST1, an enzyme catalyzing the synthesis of STs from ceramide precursors (Figure 1).

## 2. Results

### 2.1. UGT8 and GAL3ST1 Expression in Human CCA

We evaluated the gene expression of *UGT8* and *GAL3ST1* in the TCGA-CHOL cohort containing 36 CCA tissues and nine adjacent non-cancerous tissues. *UGT8* and *GAL3ST1* showed higher expression levels in tumor tissue compared to normal tissue, with significance reached for *GAL3ST1* (Figure 2A,B).

In a small cohort of patients from the Maastricht University Medical Center and Uniklinik RWTH Aachen, *UGT8* and *GAL3ST1* expression was measured by quantitative real-time PCR (qRT-PCR) in 6 tumor tissue and 10 tumor-distal liver specimens from patients with intrahepatic CCA. We found increased expression of both genes in intrahepatic CCA (iCCA) tumor specimens (Appendix A).

Next, we evaluated the expression of *UGT8* and *GAL3ST1* in CCA cell lines from different anatomical regions in the biliary tree and compared this to the expression in the normal human cholangiocyte cell line H69. This analysis revealed increased levels of *UGT8* and *GAL3ST1* in four out of five CCA cell lines (Appendix A).

The expression body map, normalized to transcripts per million (TPM) revealed the median expression of *UGT8* and *GAL3ST1* in different human cancers. *UGT8* showed values of 0.8 in CCA and 0.01 in normal tissues; however, *GAL3ST1* reached values of 70.6 in CCA and 2.3 in normal tissue (Figure 2C–F). *GAL3ST1* showed the highest expression levels in kidney cancer (label: KIRC) and CCA (label: CHOL); so, *GAL3ST1* may be involved in the pathophysiology of CCA. However, this up-regulation in CCA is not correlated with lower survival in patients with CCA (Appendix A).

### 2.2. GAL3ST1 Deficiency Reduces Cell Viability, Cell Proliferation, and Clonogenic Capacity in TFK1 Cells

We used the CRISPR-Cas9 gene editing tool to generate *GAL3ST1*-deficient cells from TFK1, an extrahepatic CCA cell line. We selected the TFK1 cell line based on the relatively high mRNA levels of *GAL3ST1* and the ability of these cells to form a monolayer, which allowed the study of the barrier function. Knockout efficiency was assessed by qRT-PCR (Figure 3A) and further supported by the quantification of cellular ST levels by liquid chromatography–mass spectrometry (LC-MS), showing 10-fold lower levels of STs in TFK1 GAL3ST1 deficient cells (Appendix A). TFK1 *GAL3ST1* deficient cells showed reduced proliferative and clonogenic capacity compared to TFK1 *wt* cells (Figure 3B,C,E).

We used the Wound Healing assay to measure the migratory capacity of the cells and their interaction. We observed that TFK1 *GAL3ST1* KO cells show a reduced migratory capacity compared to TFK1 *wt* cells (Figure 3D).

We measured the activation of ERK, one of the most studied survival pathways in CCA. ERK phosphorylation activates pathways such as Myc and ribosomal protein S6, which induce cell proliferation. Similarly, ERK activation inhibits tumor suppressor molecules [19]. TFK1 *GAL3ST1* KO cells had reduced basal levels of phospho-ERK1/2 (pERK1/2) compared to TFK1 *wt* cells (Figure 3F).

### 2.3. GAL3ST1 Deficiency Reprograms Glycolysis and Bioenergetic Metabolism in TFK1 Cells

Hypoxia is a common feature in solid tumors and enhances tumor aggressiveness by increasing glycolytic activity in tumor cells through the activation of the hypoxia-inducible factor 1α (HIF1α) pathway [20,21]. *GAL3ST1*-deficient cells displayed lower mRNA expression of *HIF1α* (Figure 4A), and glycolytic enzymes phosphoglycerate kinase 1 (*PGK1*) and hexokinase-2 (*HK2*) relative to TFK1 *wt* cells (Figure 4B,C).

To directly study the effect of *GAL3ST1* deficiency on the bioenergetic metabolism of TFK1 cells, we used the Seahorse XF Cell Mito Stress test to evaluate mitochondrial oxidative phosphorylation, measured as the oxygen consumption rate (OCR), and glycolysis, measured as the extracellular acidification rate (ECAR), which is mainly associated with lactate production.

TFK1 *GAL3ST1*^−/−^ cells showed reduced OCR values compared to TFK1 *wt* cells under basal conditions (Figure 5C); however, they showed the same relative reduction in OCR values in response to the ATP synthase inhibitor oligomycin (Figure 5A,C). Based on these results, ATP production and the membrane proton leak are not affected by the absence of *GAL3ST1* in TFK1 cells. In contrast, after the addition of the protonophore uncoupler FCCP, the maximal OCR activity was significantly lower (≈60% of TFK1 *wt* cells) in TFK1 *GAL3ST1*^−/−^ cells (Figure 5A,C). This indicates that the OCR associated with ATP synthesis is not different in TFK1 *GAL3ST1*^−/−^ cells, while the spare respiratory capacity in these cells was significantly lower than in TFK1 *wt* cells. In addition, OCR activity was similar in TFK1 *wt* and KO cells after Antimycin A/Rotenone injection (Figure 5A). Similarly, ECAR was not different between TFK1 *GAL3ST1*^−/−^ and TFK1 *wt* cells under basal conditions but was significantly reduced in the KO cells under conditions that stimulated anaerobic glycolysis (viz. oligomycin treatment) (Figure 5B). ECAR was repressed to similar levels in TFK1 *wt* and KO cells after exposure to 2-DG, a competitive inhibitor of glucose (Figure 5B). This was associated with a reduction in the glycolytic capacity and glycolytic reserve in TFK1 *GAL3ST1*^−/−^ cells (Figure 5D). The combined results indicate a lower glycolytic and mitochondrial oxidative phosphorylation capacity in TFK1 *GAL3ST1*^−/−^ relative to TFK1 *wt* cells.

### 2.4. GAL3ST1 Deficiency Improves the Barrier Function in TFK1 Cells

Cholangiocytes are polarized cells that are responsible for modifying and transporting bile released by hepatocytes. Biliary epithelial cells maintain the barrier function of the biliary tract to protect the liver parenchyma from bile toxicity [22]. To generate an in vitro model that reproduces the barrier function, TFK1 cells were grown on transwell inserts to allow the assessment of paracellular transport. Cell cultures on these porous inserts induce differentiation from nonpolarized to polarized cells. The transepithelial electrical resistance test (TEER) was used to determine the time required for the differentiation of TFK1 cells. As shown in Figure 6A, maximum TEER values for TFK1 *wt* cells were reached after 20 days in culture. The gene expression of tight junction markers zonula occludens (*ZO-1*) and occludin (*OCLN*) was significantly increased in TFK1 polarized cells on day 20 (Figure 6B). Similarly, immunofluorescence staining showed higher ZO-1 protein levels in polarized cells compared to nonpolarized cells (Figure 7B).

This monolayer cell culture model was used to evaluate the barrier function in *GAL3ST1*-deficient TFK1 cells. TFK1 *GAL3ST1*^−/−^ cells showed higher TEER values relative to TFK1 *wt* polarized cells (Figure 6C). The fluorescein isothiocyanate (FITC)-dextran permeability assay was used to assess paracellular flow from the apical to basal compartment in polarized cells. *GAL3ST1*-deficient cells showed a significant reduction in permeability compared to *wt* cells (Figure 6D). The aggregated findings indicate that *GAL3ST1* deficiency enhances the epithelial barrier function in TFK1 cells.

### 2.5. GAL3ST1 Deficiency Partially Restores Epithelial Identity in TFK1 Cells

EMT involves the loss of epithelial identity to acquire a mesenchymal phenotype that promotes cancer metastasis [23]. Metabolic reprogramming during tumor development in CCA is mediated by EMT-associated transcription factors, thereby increasing metastatic capacity [24]. We evaluated the effect of *GAL3ST1* deficiency on EMT. We analyzed the expression of epithelial markers in TFK1 *wt* cells and TFK1 *GAL3ST1^−/−^* cells. We found upregulated levels of epithelial markers *OCLN* and *CLDN1* in TFK1 *GAL3ST1*^−/−^ cells. In contrast, we did not observe a down-regulation of mesenchymal markers (Figure 7A).

In addition to elevated transcript levels (Figure 7A), we found higher levels of the ZO1 protein in TFK1 *GAL3ST1^−/−^* cells, with a notable further increase in the polarization of TFK1 cells (Figure 7B).

### 2.6. Dependency on GAL3ST1 for the Growth of Human CCA Cell Lines

To extend and validate the findings based on TFK1 cells, we used the DepMap database (https://depmap.org/portal/, accessed on 15 May 2024), which is based on CRISPR loss-of-function screens in tumor cell lines. Hereto, we used the 23Q4 public data release from the DepMap at the Broad Institute, consisting of dependency data for 17,386 genes across 1086 cancer cell lines. We analyzed dependency on *GAL3ST1* for growth in all 34 available CCA cell lines, which showed a negative score in 30 CCA cell lines (Appendix A). A more negative score means a greater cell line dependence for *GAL3ST1* (Figure 8). These data reinforce the idea of GAL3ST1 as a new therapeutic target in CCA.

## 3. Discussion

Cellular metabolic reprogramming is a hallmark of cancer, including CCA [25,26]. Glucose metabolism is deregulated in cancer cells, showing increased anaerobic glycolysis related to the aggressiveness of tumor cells. HK2, the rate-limiting enzyme involved in glycolysis, is upregulated in CCA [27]. Lipid metabolism also undergoes reprogramming in CCA, showing increased de novo lipogenesis and enhanced fatty acid uptake by tumor cells. This increase in lipid uptake makes CCA cells more mitogenic and invasive, increasing the aggressiveness and worsening the prognosis of this type of cancer [28]. STs are a group of sulfated glycosphingolipids found in different organs and tissues in the human body and have been related to the development and progression of different types of cancer [29]. The specific location of STs in the biliary tract was reported [16], and a high ratio of unsaturated to saturated STs within the tumor was correlated with earlier tumor recurrence in iCCA [17]. In the present study, we evaluated the role of GAL3ST1, the enzyme catalyzing the final step in ST synthesis after the initial UGT8-mediated step, in a CCA cell line. Our findings suggest that the inhibition of ST synthesis by the repression of GAL3ST1 could be considered a novel therapeutic target in CCA. Along this line of reasoning, zoledronic acid, approved for the treatment of osteoporosis and bone metastasis and identified as a UGT8 inhibitor, reduced the tumorigenesis of basal-like breast cancer cells in vitro [30]. To our knowledge, specific inhibitors of GAL3ST1 have not been approved [31].

*UGT8* and *GAL3ST1* showed higher mRNA levels in iCCA tumor tissue as well as in CCA cell lines from different anatomical regions of the biliary tract compared to normal tissue/cells. Based on the potential dysregulation of ST metabolism in CCA, we sought to elucidate the role of STs through the gene disruption of *GAL3ST1* in CCA cells. We generated *GAL3ST1*-deficient cells from TFK1, an extrahepatic cell line. TFK1 *GAL3ST1*-deficient cells each showed reduced tumorigenic properties, including lower proliferation, migration, and clonogenic capacity. This reduction in cell proliferation correlates with lower ERK activation, a well-known survival pathway for CCA, in *GAL3ST1*-deficient cells [19].

Glucose metabolism is dysregulated in cancer cells and is characterized by increased glucose uptake and conversion to lactate. This phenomenon of anaerobic glycolysis in the face of an adequate supply of oxygen is known as the Warburg effect [32,33]. We used the Seahorse assay to evaluate mitochondrial respiratory and glycolytic metabolism in CCA cells. TFK1 *GAL3ST1*-deficient cells produced similar amounts of ATP compared to TFK1 *wt* cells, with lower oxygen consumption and lower glycolytic capacity. In addition, *GAL3ST1*-deficient cells showed reduced mRNA levels of the glycolytic enzymes *HK2* and *PGK1*. Based on these results, STs may influence the Warburg effect in TFK1 cells and may be part of the mechanism by which tumor cell growth is induced by adapting to low oxygen levels. Hypoxia is a common condition in solid tumors, including CCA, and activates glycolysis through the activation of HIF1α [34]. In the present study, we found reduced expression of *HIF1α* in *GAL3ST1*-deficient CCA cells. This finding suggests that GAL3ST1 is involved in hypoxia in CCA by modulating the HIF1α pathway. Previously, *GAL3ST1* was identified as a HIF1α target gene in renal clear cell carcinoma [35].

The biliary tract is lined with cholangiocytes, polarized epithelial cells that modify bile through absorption and secretory processes under physiological conditions, but also form a barrier to protect liver parenchyma from toxic components in bile [36]. The integrity of the biliary tract and apical to basal polarity is established by, among others, tight junctions (TJs), which are protein complexes that connect adjacent epithelial cells [9]. We studied the barrier function of CCA cells using confluent monolayers of polarized cells. TEER was increased, and TJ-associated genes (*OCLN* and *ZO1*) were upregulated after differentiation. The cell culture system employing a permeable membrane was originally designed to obtain access to both apical and basolateral compartments in vitro. This model has been widely applied to study the barrier function in different organs, including the blood–brain barrier, gastrointestinal tract, and pulmonary epithelial barrier [37].

In the present study, we demonstrated that *GAL3ST1* deficiency enhanced the barrier function of polarized TFK1 cells. This enhancement of barrier function was reflected by increased TEER and decreased paracellular permeability in *GAL3ST1*-deficient cells. TEER values and permeability are sensitive indicators of the integrity of cellular barriers, being critical for the physiological activities of the tissue [37,38]. Loss of barrier function and leaky bile duct-induced cell injury are common mechanisms in many cholangiopathies, including PSC, PBC, and CCA [9]. In CCA, the disruption of the barrier function and TJ was associated with epithelial cell transformation, cancer cell metastasis, and invasion [39].

Based on these results and the nature of sulfatides, which are components of cell plasma membranes, we believe that cells deficient in GAL3ST1 partially recover their epithelial identity and therefore show lower proliferation, clonogenic capacity, and glycolytic and mitochondrial oxidative phosphorylation capacity.

In this regard, EMT is a process comprising modifications of gene expressions related to the suppression of the epithelial phenotype and the activation of a mesenchymal phenotype. The EMT process affects adherent junctions and the deregulation of E-cadherin, an important protein for the formation of these cell–cell junctions, which constitutes one of the key characteristics of the EMT [8]. At the same time, the disruption of tight junctions leads to a loss of cell polarity, and the acquisition of a mesenchymal phenotype enables the tumor cells to gain invasive properties [40]. Our data showed that TFK1 *GAL3ST1*-deficient cells had increased mRNA levels of E-cadherin and TJ proteins, including CLDN1, OCLN, and ZO1, compared to *wt* cells. The combination of the augmented expression of E-cadherin and various tight junction markers indicated that TFK1 *GAL3ST1*-deficient cells had enhanced epithelial identity relative to TFK1 *wt* cells [24].

We demonstrated an anti-tumor effect by deleting the *GAL3ST1* gene in TFK1 cells. We also explored the underlying mechanisms by evaluating glucose metabolism, barrier function, and phenotypes related to EMT. Further studies are needed to determine the role of GAL3ST1 in CCA tumor biology by evaluating additional cell lines and in vivo studies in experimental models of CCA. However, this study indicates that the inhibition of GAL3ST1 may have therapeutic potential in CCA by inhibiting tumor growth and aggressiveness.

## 4. Materials and Methods

### 4.1. Human Sample Collection

We collected 6 tumor tissue and 8 tumor-distal liver specimens from patients with intrahepatic CCA who underwent hepatic resection from March 2018 to January 2021 at the Maastricht University Medical Center or Uniklinik RWTH Aachen. Fresh tissue samples were snap-frozen in liquid nitrogen after excision and subsequently stored at −80 °C. All patients provided written informed consent for the use of their samples for biomedical research. This study was approved by the local medical ethics committees (Maastricht: #16-4-153, Aachen: EK 206/09).

### 4.2. Hub Gene Validation by the TCGA and GEPIA Database

RNA-seq data were extracted from the cancer genome atlas (TCGA-CHOL) using the Xena Functional Genomics Explorer (https://xenabrowser.net/, accessed on 15 May 2024) and comprised data from 36 CCA tumor samples. Gene Expression Profiling Interactive Analysis (GEPIA; http://gepia.cancer-pku.cn/, accessed on 15 May 2024), an online tool based on the Cancer Genome Atlas (TCGA) and Genotype Tissue Expression (GTEx) databases, was used to visualize and study gene expression levels between tumor and normal tissues and to assess the prognostic value in CCA [41].

### 4.3. Cell Culture

Human extrahepatic CCA cell lines (TFK1 and EGI-1), intrahepatic CCA cell lines (CC-LP1 and SNU1079), gallbladder carcinoma cell line (MzCha1), and a normal human cholangiocyte cell line (H69) were kindly provided by Prof. Matias Avila, University of Navarra, Spain. CCA cell lines were cultured in DMEM/F-12 (Gibco, Waltham, MA, USA), supplemented with 10% (*v*/*v*) fetal bovine serum (FBS) and 1% (*v*/*v*) penicillin/streptomycin (P/S) (both from Thermo Scientific, Waltham, MA, USA). All cell lines were cultured at 37 °C under 5% CO_2_ in a humidified incubator. The cells were tested for mycoplasma with the MycoAlert Mycoplasma kit (Lonza, Basel, Switzerland) regularly during experiments and were found to be negative.

### 4.4. Knockout of GAL3ST1 Genes by CRISPR-Cas9 in TFK1 Cells

TFK1 cells deficient in *GAL3ST*1 were generated by the CRISPRi-Cas9 technology. Briefly, TFK1 cells were seeded into 6 well plates at a density of 0.05 × 10^6^ cells per well and cultured in an antibiotic-free medium until 60% confluency. Then, the cells were co-transfected with a CRISPRi vector containing Cas9 and a mixture of three single-guide RNAs (sgRNA) to target *GAL3ST1* genes (sc-407936; Santa Cruz Biotechnology, Dallas, TX, USA) and a homology-directed repair (HDR) vector containing a puromycin resistance gene flanked by two LoxP sites (sc-407936HDR; Santa Cruz Biotechnology, Dallas, TX, USA) for 72 h. TFK1 *wt* cells were generated using CRISPR/Cas9 Control plasmid (sc-418922; Santa Cruz Biotechnology, Dallas, TX, USA). The cells were subsequently cultured in the presence of 0.5 µg/mL puromycin (sc-108071, Santa Cruz Biotechnology, Dallas, TX, USA) for 5 days to select transfected cells. After selection, the different assays were carried out, with cells maintained in a complete medium.

### 4.5. MTS Proliferation Assay

The cell proliferation and viability were monitored with an MTS assay according to the manufacturer’s protocol. Briefly, the cells were seeded in a 96-well plate at a density of 0.05 × 10^6^ cells per well. The cells were cultured in 100 µL of culture medium for 72 h after the seeding, and 20 µL of CellTiter 96^®^ AQueous One Solution Reagent (Promega, Madison, WI, USA) was added to each well. Next, the plate was incubated at 37 °C for 1 h. The absorbance was measured at an optical density (OD) value of 490 nm with a microplate reader (TECAN SPARK 10M, Männedorf, Switzerland).

### 4.6. Live-Cell Proliferation Assay

The cells were plated in a 96-well plate at a density of 0.01 × 10^6^ cells per well and placed in an IncuCyte^®^ S3 (Sartorius, Göttingen, Germany) inside a humidified incubator operating at 37 °C in a 5% CO_2_ atmosphere. Time-lapse images were acquired with a 10× objective. Stacks of images were exported in a tagged image file (TIF) format for cell morphology analysis. The data from real-time analysis of the occupied area (% confluency) were exported to Excel for proliferation analysis.

### 4.7. Colony Formation Assay

The cells were seeded into a 6-well plate at a density of 500 cells per well and cultured for 14 days to produce colonies. The cell culture medium was refreshed every two days. The colonies were washed with PBS (Gibco, Waltham, MA, USA), fixed with ice-cold methanol (Sigma-Aldrich, St. Louis, MO, USA), and stained with 0.5% (*w*/*v*) crystal violet (Sigma-Aldrich, St. Louis, MO, USA). The growth area of the colonies was analyzed using Image J.

### 4.8. Cell Homogenate Preparation, Sulfatide Extraction, and Mass Spectrometry Analysis

After 24 h in culture, cell pellets of 3.105 cells were collected by scraping and homogenized in 150 μL of sterile PBS. Sulfatide content was quantified by liquid chromatography–mass spectrometry (LC-MS-MS) as previously described [42].

### 4.9. Live-Cell Metabolic Analysis

Mitochondrial and glycolytic functionality in cells was assessed by a Seahorse XF-96 analyzer (Seahorse Biosciences, Billerica, MA, USA) using the Seahorse XF Cell Mito stress test kit (Agilent, Santa Clara, CA, USA) and the Seahorse XF Glycolysis stress test kit (Agilent, Santa Clara, CA, USA), respectively. Cell density and test compound concentrations were optimized in pilot experiments. An optimized number of cells (TFK1 *wt* 30 × 10^3^ cells/well and TFK1 *GAL3ST1* KO 15 × 10^3^ cells/well) were seeded into a Seahorse XF-96 cell culture plate (Agilent, Santa Clara, CA, USA) and incubated overnight in growth media. For the Mito stress assay, we used DMEM D5030 (Sigma-Aldrich, St. Louis, MO, USA) supplemented with 143 mM NaCl (Sigma-Aldrich, St. Louis, MO, USA), 1× Glutamax (Gibco, Waltham, MA, USA), 0.1 mM sodium pyruvate (Sigma-Aldrich, St. Louis, MO, USA), 3 mg/L phenol red (Sigma-Aldrich, St. Louis, MO, USA), and 25 mM glucose (Sigma-Aldrich, St. Louis, MO, USA). For the glycolysis stress test, we used DMEM D5030 (Sigma-Aldrich, St. Louis, MO, USA) supplemented with 143 mM NaCl (Sigma-Aldrich, St. Louis, MO, USA), 1× Glutamax (Gibco, Waltham, MA, USA), and 2 mM L-glutamine (Sigma-Aldrich, St. Louis, MO, USA). The cells were placed in an incubator without CO_2_. The medium was refreshed before the tests.

For the Mito Stress test, the oxygen consumption rate (OCR) was recorded before and after the addition of the following inhibitors (Mito Stress Test Kit; Agilent, Santa Clara, CA, USA): 1 μM oligomycin A (A), 1.5 mM 4-phenylhydrazone (FCCP) (B), and 1 μM antimycin A + 1 μM rotenone (C). Oligomycin was used to inhibit ATP synthesis, FCCP injection was used to determine maximal respiration, and antimycin A/rotenone was used to measure the spare respiratory capacity. For the glycolysis stress test, the results were reported as the extracellular acidification rate (ECAR). Glycolytic flux (baseline of glycolysis, glycolytic capacity, and glycolytic reserve) was measured before and after the addition of the following compounds: 10 mM glucose, 1.5 μM oligomycin, and 100 mM 2-deoxyglucose (2-DG, glycolysis inhibitor). All the results were normalized to cellular protein assessed by Rapid Gold BCA Protein Assay Kit (Thermo Scientific, Waltham, MA, USA).

### 4.10. Cell Differentiation, Transepithelial Electrical Resistance Assay, and Permeability Assay

The cells were seeded at a density of 0.1 × 10^6^ cells onto Millicell^®^ hanging cell culture inserts (0.4 µm PET) (Sigma-Aldrich, St. Louis, MO, USA) in a 24-well plate and grown for up to 20 days. The upper and lower chambers of the inserts contained 200 µL and 700 µL of media, respectively. The integrity of the cell barrier was determined by transepithelial electrical resistance (TEER) measurement and a permeability assay. For the TEER assay, the electrical resistance was measured using an EVOM2 Epithelial Volt/Ohm Meter (World Precision Instruments, Sarasota, FL, USA) before changing the medium. The TEER value of the cell layer was determined by subtracting the background resistance of the semipermeable membrane without cells and multiplying it with the cell culture surface area; it is reported as Ω.cm^2^.

After the final TEER measurements, the permeability of the cell barrier was assessed by measuring the paracellular passage of fluorescein isothiocyanate-labeled dextran (4 kDa, FITC-D4) (Sigma-Aldrich, St. Louis, MO, USA). The cell culture inserts were placed into a new 24-well plate, and the apical medium was replaced by a medium containing 1 mg/mL of FITC-dextran 4 kDa, while the basal medium was replaced by PBS. Aliquots were withdrawn from the luminal and basal compartments after one hour of incubation at 37 °C, and fluorescence intensity was determined at 485 nm excitation and 530 nm emission with a microplate reader (TECAN SPARK 10M, Männedorf, Switzerland). Data are expressed as relative fluorescence units.

### 4.11. Immunofluorescence Staining

Immunofluorescence staining was performed on cells grown in polarized and nonpolarized ways for the analysis of the expression of the tight junction marker ZO-1. The polarized cells used for immunofluorescence staining were the same as those grown on the hanging cell culture inserts for the permeability assay, while the nonpolarized cells were grown on the culture insert 4 well in μ-dish 35 mm (Ibidi, Gräfelfing, Germany) for 4 days. The cells were fixed with 4% paraformaldehyde (Sigma-Aldrich, St. Louis, MO, USA) for 30 min, permeabilized with 0.5% Triton X-100 (Sigma-Aldrich, St. Louis, MO, USA) for 30 min, and blocked with 1% BSA overnight. The cells were incubated with mouse monoclonal ZO-1 antibody-FITC (cat no. 339111, Thermo Scientific, Waltham, MA, USA). The cell nuclei were counterstained with blue Hoechst 33,258 (cat no. H3569, Thermo Scientific, Waltham, MA, USA). Finally, the cells were mounted with VECTASHIELD^®^ Antifade Mounting Medium with DAPI (Dako-Agilent, Glostrup, Denmark). The images were captured with a confocal microscope from Leica Microsystem (Wetzlar, Germany).

### 4.12. RNA Isolation, cDNA Synthesis, and Quantitative Real-Time PCR Analysis

Total RNA was extracted using TRI Reagent (Sigma-Aldrich, St. Louis, MO, USA), and cDNA was synthesized using the Sensifast cDNA Synthesis Kit (Bioline GmbH, Luckenwalde, Germany). Quantitative real-time PCR (qRT-PCR) analysis was performed on a LightCycler480 system (Roche, Mannheim, Germany). The expression values were calculated using LinRegPCR software and expressed relative to the reference gene *36B4*. Specific primer pairs for each gene were designed using PrimerBlast and purchased from Sigma-Aldrich (St. Louis, MO, USA); the sequences are listed in Appendix A. The amplicon sizes were verified by the agarose gel electrophoresis of PCR products.

### 4.13. Gene Dependency Data in CCA Cell Lines

Public cancer dependency data (https://depmap.org/portal/, accessed on 15 May 2024) were used to evaluate the *GAL3ST1* dependency of different CCA cell lines. This database consists of a comprehensive set of genes knocked out using the CRISPR-Cas9 technology in a broad panel of transformed human cell lines. The difference in the effect of *GAL3ST1* targeting on distinct CCA cell lines was calculated as T-statistic scores using scipy.stats.ttest_ind (available via the DepMap portal). The results are presented as CRISPR scores, showing the dependency of each cell line on *GAL3ST1*. Negative scores show an influence of the studied gene on cell proliferation and viability.

### 4.14. Western Immunoblotting

The cell lysates were prepared in a radioimmunoprecipitation assay (RIPA) buffer supplemented with cocktails of phosphatase (cat no. 4906845001, Roche, Basel, Switzerland) and protease inhibitors (cat no. P8215, Sigma-Aldrich, St. Louis, MO, USA). The protein concentrations were quantified by Rapid Gold BCA Protein Assay Kit (Thermo Scientific, Waltham, MA, USA) according to the manufacturer’s protocol. The western immunoblotting assay was performed as described previously [43]. Briefly, 30 μg protein in Laemmli buffer (Sigma-Aldrich, St. Louis, MO, USA) was separated on 4–20% polyacrylamide gel (BioRad, Hercules, CA, USA) and transferred to polyvinylidene fluoride (PVDF) membranes. The membranes were then blocked with 5% *w*/*v* fat-free dry milk (Sigma-Aldrich, St. Louis, MO, USA) and incubated with primary antibodies (anti-phospho-p44/42 MAP kinase (ERK1/2) or anti-ERK) (Cell Signaling, Danvers, MA, USA) overnight at 4 °C, rinsed, and incubated with HRP-conjugated secondary antibodies for one hour at room temperature. Peroxidase activity was visualized using SuperSignal™ West Pico chemiluminescent substrate (Thermo Scientific, Waltham, MA, USA). The images were obtained with a molecular imager, namely, the Amersham Imager 600 (GE Healthcare Life Sciences, Marlborough, MA, USA). The protein bands were quantified by Image J.

### 4.15. Wound Healing Assay

The cells were plated at 100% of confluence and subsequently starved for 16 h. A wound was created in the cell monolayer using a sterile P10 pipette tip. After inflicting the wound, the cells were washed twice with PBS and then cultured in a normal culture medium. Imaging was conducted with an IncuCyte^®^ S3 (Sartorius, Göttingen, Germany) at the initial time (0 h) and subsequently at 24 and 48 h after scratching. The areas of the wound were quantified using Image J software equipped with the MRI Wound Healing tool. The percentage of the wound opening was calculated with the formula: (wound area at time t/initial wound area) × 100%.

### 4.16. Statistical Analysis

The non-parametric Mann–Whitney U test (for two groups) and Kruskal–Wallis test (for three or more groups, followed by Dunn’s multiple comparison test as a post-hoc test) were used for statistical comparisons. *p*-values < 0.05 were considered significant. We report *p*-values with asterisks: * *p* < 0.05, ** *p* < 0.01, and *** *p* < 0.001. Statistical analyses were performed using GraphPad Prism ver. 7.01 software (San Diego, CA, USA) for Windows. All in vitro experiments were replicated at least three times. The numerical results from our experiments are expressed as mean ± standard deviation (SD), whereas data extracted from the database are presented as median and range.

## 5. Conclusions

The alteration of sulfatide metabolism in TFK1 cells recovered an epithelial identity and barrier function that may be related to reduced tumorigenic activity and reprogramming of glucose metabolism in the TFK1 cell line. Inhibiting sulfatide synthesis by targeting GAL3ST1 holds potential as a new therapeutic strategy in cholangiocarcinoma.

## Figures and Tables

**Figure 1 ijms-25-07279-f001:**
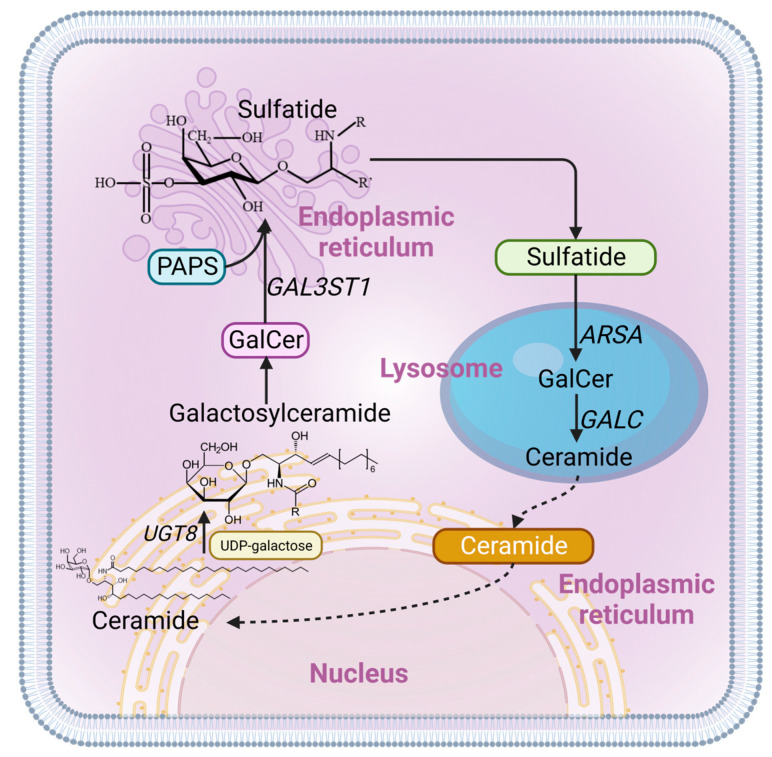
Schematic overview of sulfatide metabolism. Sulfatide synthesis begins in the endoplasmic reticulum with the addition of galactose from UDP-galactose (UDP-Gal) to ceramide by the enzyme ceramide galactosyltransferase (CGT)/UGT8, forming galactosylceramide (GalCer). GalCer is then transported to the Golgi apparatus, where 3′O-sulfation of galactose takes place, catalyzed by cerebroside sulfotransferase (CST)/GAL3ST1. Sulfatide degradation takes place in the lysosomes, where arylsulfatase A (ARSA) removes the sulfate group and galactocerebrosidase (GALC) hydrolyzes GalCer to ceramide. A complete cycle of sulfatide metabolism is depicted with solid lines, while the beginning of synthesis and the end of degradation, marked by the recycling transport of ceramide, are illustrated with dashed lines.

**Figure 2 ijms-25-07279-f002:**
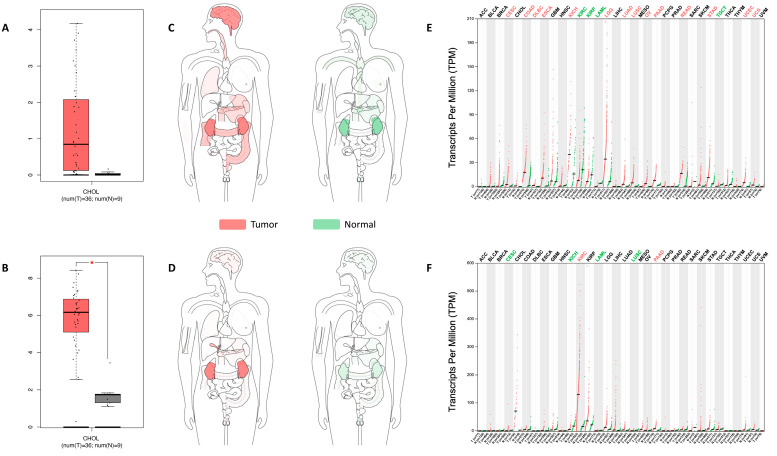
*UGT8* and *GAL3ST1* expression in human iCCA tumor specimens. *UGT8* (**A**) and *GAL3ST1* (**B**) expression in unclassified CCA tissue (T) (*n* = 36) compared to non-malignant liver tissue (N) (*n* = 9) from the Cancer Genome Atlas human database. Data are shown as the median with the range; * *p* < 0.05. *UGT8* (**C**) and *GAL3ST1* (**D**) median expression of tumor (red) and normal tissue (green) in the body map from the GEPIA (Gene Expression Profiling Interactive Analysis) human database. *UGT8* (**E**) and *GAL3ST1* (**F**) expression profiles across all tumors (red) and paired normal (green) tissues from the GEPIA database. Each dot represents the expression in one sample.

**Figure 3 ijms-25-07279-f003:**
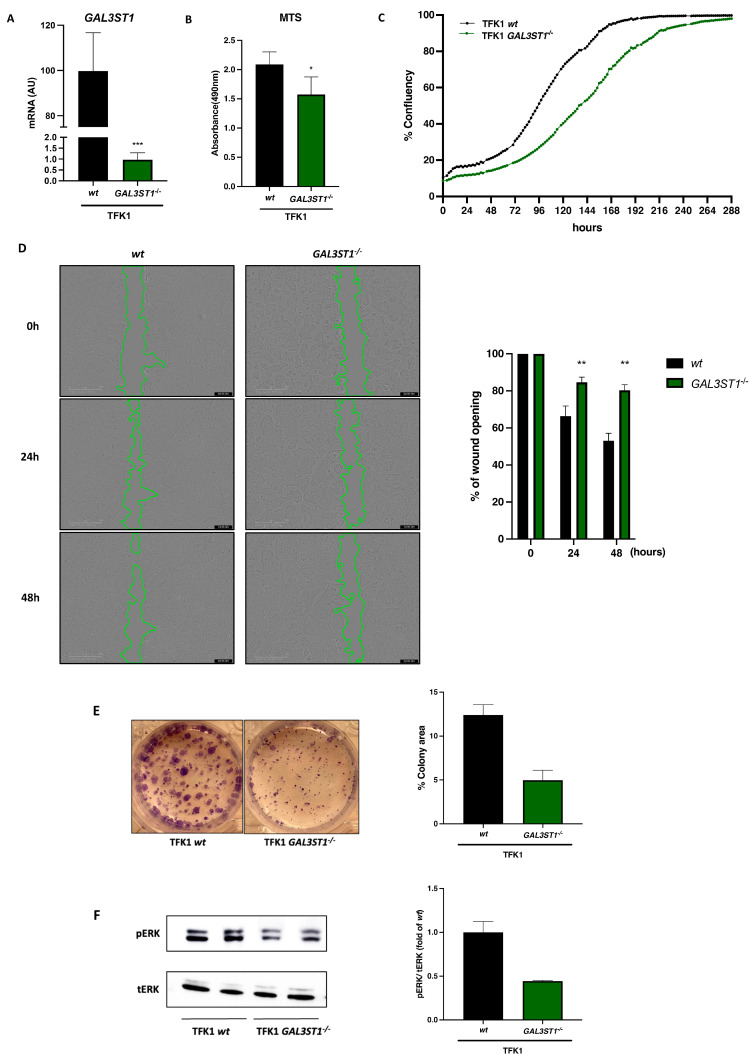
Effect of *GAL3ST1* deficiency on cell proliferation of TFK1 cells. (**A**) Relative mRNA levels of *GAL3ST1* and (**B**) MTS assay in TFK1 *wt* and *GAL3ST1*^−/−^ cells; *n* = 4. (**C**) Cell proliferation analysis of TFK1 *wt* and *GAL3ST1*^−/−^ cells assessed by live-cell imaging; *n* = 10. (**D**) Wound Healing assay in TFK1 *wt* and *GAL3ST1*^−/−^ cells; *n* = 6. (**E**) Colony formation assay of TFK1 *wt* and *GAL3ST1*^−/−^ cells; *n* = 2. (**F**) Western blot analysis of phospho-ERK1/2 (*p*-ERK1/2) and total ERK (tERK) in TFK1 *wt* and *GAL3ST1*^−/−^ cells. Data are shown as mean ± SD. * *p* < 0.05, ** *p* < 0.01, and *** *p* < 0.001 vs. TFK1 *wt* cell.

**Figure 4 ijms-25-07279-f004:**
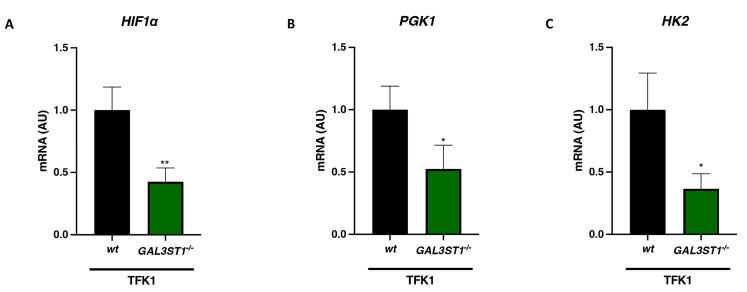
Effect of *GAL3ST1* deficiency on the expression of hypoxia-associated glycolytic genes in TFK1 cells. Relative mRNA levels of hypoxia-inducible factor *HIF1α* (**A**) and the glycolytic enzymes *PGK1* (**B**) and *HK2* (**C**) in TFK1 *wt* and *GAL3ST1*^−/−^ cells; *n* = 4. Data are shown as mean ± SD. * *p* < 0.05 and ** *p* < 0.01 vs. TFK1 *wt* cells.

**Figure 5 ijms-25-07279-f005:**
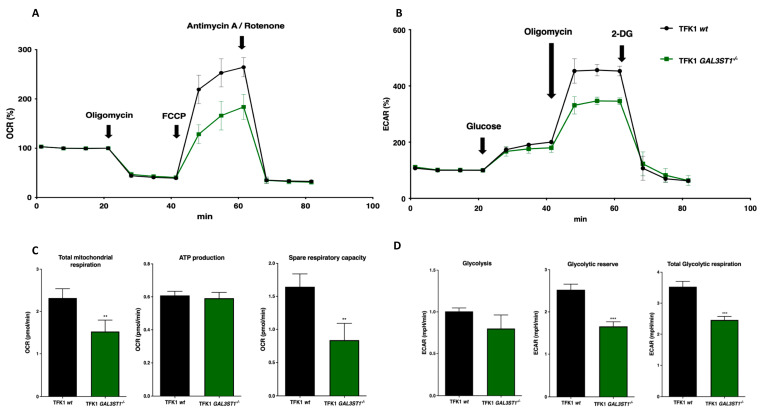
Effect of *GAL3ST1* deficiency on the metabolic fluxes in TFK1 cells. (**A**) Oxygen consumption rate (OCR) and (**B**) extracellular acidification rate (ECAR) in TFK1 *wt* and *GAL3ST1*^−/−^ cells. (**C**,**D**) Metabolic parameters inferred from the ECAR and OCR assays. *n* = 4 per condition. Data are shown as mean ± SD. ** *p* < 0.01 and *** *p* < 0.001 vs. TFK1 *wt* cells.

**Figure 6 ijms-25-07279-f006:**
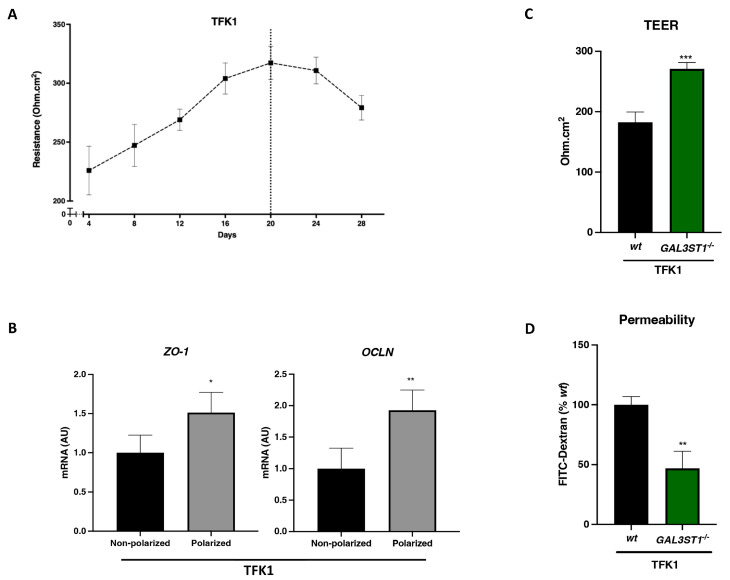
*GAL3ST1* deficiency improves barrier function in polarized TFK1 cells. (**A**) Transepithelial electrical resistance (TEER) values during the polarization process in TFK1 *wt* cells; (**B**) relative mRNA levels of tight junction markers *ZO-1* and *OCLN* in TFK1 polarized cells cultured for 20 days; (**C**) TEER values at day 20 of the differentiation process in TFK1 *wt* and *GAL3ST1*^−/−^ cells; and (**D**) FITC-dextran fluorescence intensity in basolateral medium (expressed relative to *wt* cells). *n* = 4. Data are shown as mean ± SD. * *p* < 0.05, ** *p* < 0.01, and *** *p* < 0.001 vs. TFK1 *wt* cells.

**Figure 7 ijms-25-07279-f007:**
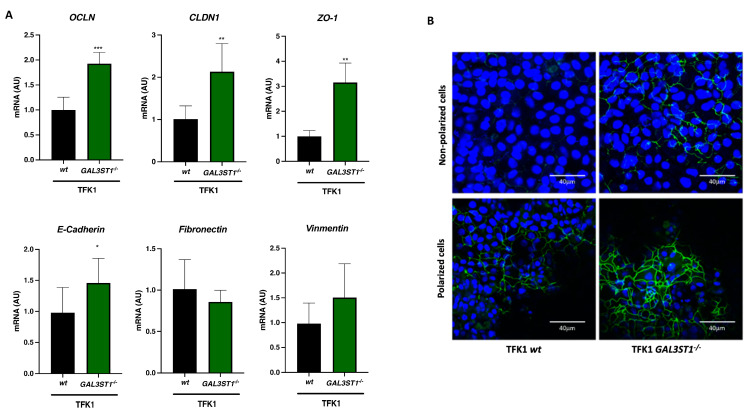
Effect of *GAL3ST1* deficiency on epithelial–mesenchymal transition (EMT) in TFK1 cells. Relative mRNA levels of (**A**) epithelial (*OCLN*, *CLDN1*, *ZO-1*, and *E-Cadherin*) and mesenchymal (*fibronectin* and *vimentin*) markers in TFK1 *wt* and *GAL3ST1*^−/−^ cells and (**B**) anti-ZO-1 immunofluorescent staining of nonpolarized and polarized TFK1 *wt* and *GAL3ST1*^−/−^ cells; *n* = 4. Data are shown as mean ± SD. * *p* < 0.05, ** *p* < 0.01 and *** *p* < 0.001 vs. TFK1 *wt* cells.

**Figure 8 ijms-25-07279-f008:**
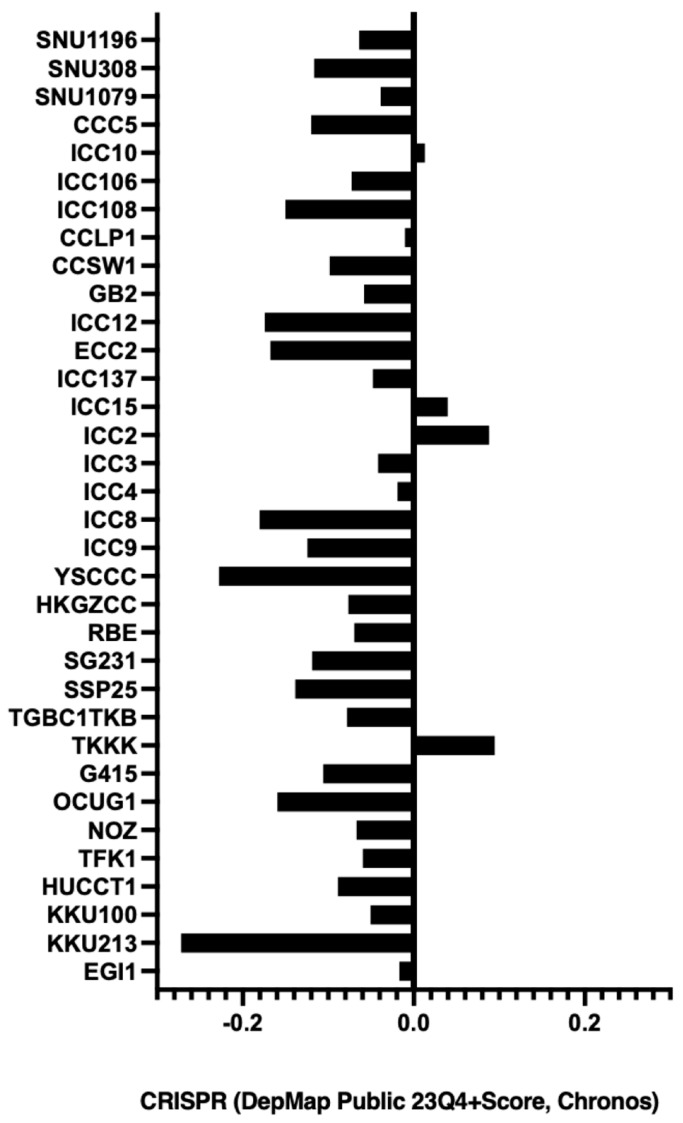
*GAL3ST1* dependency for cell survival and proliferation in 34 human CCA cell lines. CRISPR screen in tumor cell lines from the 23Q4 public data release from the Cancer dependency map portal. Negative DepMap scores indicate *GAL3ST1* dependency for cell survival and proliferation.

## Data Availability

Data supporting the findings of this study are available from the corresponding author upon reasonable request.

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
