# Peer review of "GAL3ST1 Deficiency Reduces Epithelial–Mesenchymal Transition and Tumorigenic Capacity in a Cholangiocarcinoma Cell Line"

_ijms, 2024, doi:10.3390/ijms25137279_

Round 1
Reviewer 1 Report
Comments and Suggestions for Authors
The submitted manuscript entitled "GAL3ST1 Deficiency Reduces Epithelial-Mesenchymal Transition and Tumorigenic Capacity in a Cholangiocarcinoma Cell Line" by Lin Chen et al. elucidates the loss of function of GAL3ST1 in TFK-1 extrahepatic cholangiocarcinoma (CCA) cells using an in vitro study. The study demonstrates a reduction in invasion capacity, which correlates with metastasis in vivo. Despite the enhanced expression of GAL3ST1 in human CCA, this gene has not been studied in CCA, making this gene a novel target. However, the overall data appear underdeveloped and overly translated.
Please see my suggestions below:
- The authors chose the TFK-1 cell line due to its most enhanced expression. However, one cell line is insufficient to support overall claims, as CCA is molecularly and genetically heterogeneous. Additionally, PSC-driven CCA is most likely intrahepatic multifocal mass-forming type. Furthermore, the current study does not contain any in vivo data, which is critical. I recommend exploring more cell lines at the very least.
- In the TCGA analysis in Figure 1, is there any correlation with survival?
- A gain-of-function study with overexpression, particularly in cell lines with lower expression, is recommended.
- Please assess proliferation and cell death in GAL3ST1 KO cell.
- What is the rationale for examining p-ERK, and what is the translational significance?
- In Figure 1B, confluency appears to plateau at later time points. Is this delayed proliferation?
- Please include a migration assay such as a scratch assay.
Reviewer 2 Report
Comments and Suggestions for Authors
In order to enhance the calibre of your research work on "GAL3ST1 Deficiency Reduces Epithelial-Mesenchymal Transition and Tumorigenic Capacity in a Cholangiocarcinoma Cell Line," it is advised to carry out the following additional corrections:
Commets
1. Perform migration and invasion evaluates in order to examine the function of GAL3ST1 in cellular movement and the ability to migrate to other parts of the body.
2. Examine the expression of a broader range of EMT indicators, including both epithelial (such as E-cadherin) and mesenchymal (such as N-cadherin, vimentin) markers.
3. Count the colonies in figure 3C and perform densitometry analysis in figure 3D.
4. The protein levels of the identified genes should be shown in Figure 4 in relation to their mRNA levels.
5. The authors should additionally conduct further cell proliferation experiments and do cell cycle analysis.
Comments on the Quality of English Language
no any
